# ADAPTIVE DISTANCE MESSAGE PASSING FROM THE MULTI-RELATIONAL EDGE VIEW

**Zhongtian Sun[1], Alexandra I. Cristea[1], Pietro Lio[2], Jialin Yu[3]**
[1]Durham University {`zhongtian.sun, alexandra.i.cristea`}`@durham.ac.uk`
[2]University of Cambridge `pl219@cam.ac.uk`
[3]University College London `jialin.yu@ucl.ac.uk`

## ABSTRACT

Message-passing graph neural networks (MP-GNNs) excel in deep learning on graphs. Despite their success in various studies, they are limited by passing information to the fixed length $k$ distance neighbouring nodes, where $k$ is the number of layers. In reality, different types of edges (alternatively relations) may influence nodes at varying distance and should not be uniformly treated. This paper proposes an adaptive distance message-passing method that considers the unique roles of edge types, addressing this issue. Experiments on real-world datasets validate the effectiveness of our approach.

## 1 INTRODUCTION

Message-passing based graph neural networks (MP-GNNs) have proven to be effective for graph representation learning (Sanchez-Lengeling et al., 2021; Bodnar et al., 2021), which aggregate the information from the neighbouring nodes iteratively and update the representation of nodes based on passed information and their previous states (Gilmer et al., 2017).

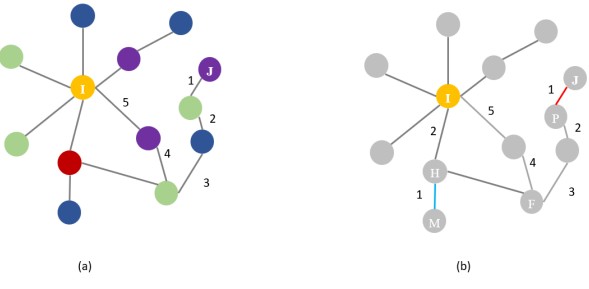

Figure 1: Local aggregation

However, MP-GNNs are limited by local message passing (He et al., 2022). Specifically, to aggregate information from $k - hop$ neighboring nodes, the neural network must stack $k$ layers, a process we refer to as *uniform length message passing*. As illustrated in Figure 1 (a), we require five layers to pass information from node $J$ to node $I$, ensuring that all nodes in the graph can capture information from $5 - hop$ neighbouring nodes. However, when more than four layers are stacked, MP-GNNs loses their generalisation ability and overly compressed information leads to over-squashing (Alon & Yahav, 2020). Although advanced methods such as non-local update mechanisms(Balcilar et al., 2021) or transformers (Ying et al., 2021; Mialon et al., 2021; Thölke & De Fabritiis, 2022) have recently been applied to MP-GNNs to capture long-range dependencies in the graph(Min et al., 2022), they tend to focus more on the influence of nodes rather than the impact of multiple types of edges over nodes. This approach is not optimal for tasks like molecule learning, where different types of relations exist. Other studies (Vashishth et al., 2019; Zheng et al., 2022) investigated message passing on multi-relational graphs but were still limited to *uniform length message passing*.

Addressing the above problems, we propose a novel method called adaptive distance message passing from edge view (ADMPE), where edges with significant impacts can reach distant nodes rather than being limited to fixed $k - hop$ neighbouring nodes in *uniform length message passing*. As illustrated in Figure 1 (b), the red edge between source node $J$ and target node $P$ can reach the five hop node $I$ while the blue edge from $M$ to $H$ reaches two-hop nodes $I$ and $F$.

## 2 METHODOLOGY

We examine the edge attributes to determine the extent of information propagation. For instance, there are four types of edges (0,1,2,3) between atoms (nodes) in the ZINC dataset, representing *no bond*, *single*, *double* and *aromatic bonds*. These bonds differ in reactivity and stability, with double bonds being more reactive than single bonds and aromatic bonds being more stable due to higher electron density. Our mechanism allows the type 1 edges to propagate to $k_1 - hop$, and type 2 edges to $k_2 - hop$ of nodes. $k_1$ and $k_2$ are distinct, learnable parameters for different edge types, initialised as integers between 1 and 10. While following standard MP-GNN steps (Gilmer et al., 2017): initialisation, aggregation, update and readout, we modify the aggregation to incorporate features of different edge types to nodes at varying hops:

$$\mathbf{h}'_i = f_u(\mathbf{h}_i, \sum_{j \in N(i)} \mathbf{W}_1 \mathbf{x}_j \sum_{i \in M(r)} \mathbf{W}_2 \mathbf{e}^r_{jp}) \tag{1}$$

$\mathbf{h}'_i$ and $\mathbf{h}_i$ represent the updated representation and previous state of node $i$, respectively. $\mathbf{x}_j$ denotes the initial features for neighbour node $j$ and $\mathbf{e}_r$ represents edge feature of relation $r$. $f_u$ is the update function and $\mathbf{W}_1$, $\mathbf{W}_2$ are learnable matrices. $N(i)$ denotes the neighbours of node $i$ and $M(r)$ is a hop count set indicating that node $i$ is within the maximum hop length $k - hop$ of a type of $r$ edge which connected with node $j$ and $p$, as shown in Figure 1 (b).

## 3 EXPERIMENT

**Datasets** We consider two benchmark datasets ZINC (12K) (Dwivedi et al., 2020)and MolHIV (Hu et al., 2020) for molecule learning tasks, details shown in Table 2 in Appendix.

**Baselines and Results.** We compare GPS(Rampášek et al., 2022), state of the art on Zinc (12K) dataset and Graph MLP-Mixer(He et al., 2022) state of the art for MolHIV, with our method ADMPE. We keep the same implementations as the baselines and integrate them with the updated function in equation 1. Table 1 summarises the mean results of four runs, demonstrating the effectiveness of ADMPE. While ADMPE's integration with GPS on the ZINC dataset shows modest improvements, its use with the Graph MLP-Mixer method noticeably enhances performance on both the ZINC and MolHIV datasets.

Table 1: Comparison of our model to baseline methods

| Methods | ZINC | MolHIV |
|---|---|---|
| Metric | MAE | ROCAUC |
| GPS | 0.070 | 0.781 |
| Graph MLP-Mixer | 0.075 | 0.807 |
| GPS with ADMPE | **0.069** | 0.785 |
| Graph MLP-Mixer with ADMPE | 0.072 | **0.811** |

## 4 CONCLUSION

This paper introduces a novel adaptive distance message-passing (ADMPE) method, focusing on multi-relational edges. ADMPE overcomes *uniform length message passing* limitations by proposing a relation-aware rule for passing edge information to nodes at varying distances. Easily integrated with existing work, ADMPE enhances representation learning. However, it requires prior knowledge of edge types in the dataset, which can be investigated in future.

## URM STATEMENT

The authors acknowledge that at least one key author of this work meets the URM criteria of ICLR 2023 Tiny Papers Track.

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

# A APPENDIX

## A.1 DATASETS

ZINC (12K) dataset consists of 28 types of nodes (atom types) and 4 types of edges (but type 0 represents *no bond* between atoms, so only 3 types of actual edges in the molecule graph). For Molhiv, edges represent chemical bonds, and nodes refer to atoms. The default setting of Molhiv in the Graph MLP-Mixer model is no types of nodes and edges. However, when loading the Molhiv dataset using ogb library, the edge attribute still has four types of values (0,1,2,3). We apply the node and edge linear embeddings (hidden dimension is 128) to the raw input, same as (He et al., 2022) and train the model.

Table 2: Dataset summary

| Datasets | Graphs | Nodes | Avg Nodes | Avg Edges | Node Types | Edge Types | Class |
|----------|--------|-------|-----------|-----------|------------|------------|-------|
| ZINC | 12000 | 9-37 | 23.2 | 24.9 | 28 | 4 | Regression (1) |
| MolHIV | 41127 | 2-222 | 25.5 | 54.9 | None | None | 2 |

## A.2 EXPERIMENTAL SETTINGS.

Our model is implemented with Pytorch 1.12.1 and CUDA 11.3. We keep the same hyper-parameters setting as baselines Graph MLP-Mixer (He et al., 2022) and GPS Rampášek et al. (2022). The whole model is implemented on PyTorch Geometric.

