# OpenReview forum: "Adaptive Distance Message Passing From the Multi-Relational Edge View"
_ICLR.cc/2023/TinyPapers — Submitted to Tiny Papers @ ICLR 2023_

### Official Review · Reviewer_ETrF · 2023-03-25

**Confidence:** 5

**Summary Of Contributions:**

The paper introduces an adaptive distance message passing scheme for graph neural networks. It aims to capture the types of edges in the graph by adapting how far the messages from different types of edges travel. The idea is applied on two different datasets and results are presented.

**Rating:**

Clear, Correct, and Reproducible (CCR): a submission which meets the reviewing criteria

**Strengths And Weaknesses:**

Strengths:
The paper states the proposed idea clearly and effectively communicates it. The motivation for the adaptive distance message passing scheme is well presented.

Weaknesses:
It is not clear whether the proposed methodology depends on the dataset in question. Since the number of edge-types is a factor, how do we use ADMPE if edge-type information is unknown or unclear? How much domain knowledge is necessary?

**Suggested Changes:**

1. Does the methodology have to be updated depending on the learning task? Specifically, do we have to know how many types of edges are there in the dataset ahead of time?
2. How many times were the experiments run? Is only the mean value presented in Table 1?

---

### Official Review · Reviewer_sqwj · 2023-03-29

**Confidence:** 3

**Summary Of Contributions:**

Authors propose a novel aggregation scheme, i.e., adaptive distance message passing aggregation, to addresses thr influence induced by the different types of edges in the graphs. Experiments on real-world datasets validate the effectiveness of our method.

**Rating:**

Great Start (GS): a submission which meets some of the reviewing criteria but has room for improvement

**Strengths And Weaknesses:**

[Strength]:
- The motivation is clear, i.e., addressing the the negative impact brought by different types of edges in a graph.

[Weakness]:
- The solution is not well stated, e.g., how to select the value of k+n-hop to perform the adaptive distance message passing? Is n a learnable parameter? I did not find where it state how to learn this paremeter.
- The experimental results seem not to be very significant, which just improve 1/1000 on zinc.


**Suggested Changes:**

- The solution need to be polished and the comparison experiment need to be redesigned.

---

### Comment · Area_Chair_w21E · 2023-06-02
**Archival**

This work meets the threshold for archival, contents the URM statement and is deanonymized

---

### Meta-Review · Area_Chair_w21E · 2023-04-04

**Recommendation:** Invite to archive
**Confidence:** 4

**Metareview:**

The paper proposes a message passing scheme where the distance over which messages are passed is dynamic. Experiments on real world datasets are presented.

**Summary:**

The reviewers agree that there is potential in the proposed method

**Comments And Feedback To The Authors:**

Please address some of the questions posed in the reviews like "is n a learnable parameter".

**Reason For Not Giving A Higher Recommendation:**

There are some crucial questions that should be answered by the authors that can bolster the claims presented in the paper.

**Reason For Not Giving A Lower Recommendation:**

Although some concerns are mentioned in the reviews, the reviewers generally agree that the paper is clear and reproducible. The concern that the results do not improve the state of the art is not applicable here.

---

### Decision · Program_Chairs · 2023-04-07

Invite to archive